# Age differences in motivation drive performance during the sustained attention to response task

Simon Hanzal[1,*], Gemma Learmonth[1,2], Gregor Thut[1,3], Monika Harvey[1]

**1** School of Psychology and Neuroscience University of Glasgow, Glasgow, United Kingdom, **2** Division of Psychology, University of Stirling, Stirling, United Kingdom, **3** The Brain and Cognition Research Centre (CerCo), CNRS and University of Toulouse, Toulouse, France

\* hanzalsimon@gmail.com

## Abstract

Young and older adults prioritise speed and accuracy differently during sustained attention tasks. While older adults generally show a preference of accuracy over speed, this is not always the case. The underlying factor behind this inconsistency may be motivational differences, with older participants compensating for a speed disadvantage with increased intrinsic motivation to perform well. We investigated this in a pre-registered study, using the Sustained Attention to Response Task (SART) in young (n = 25, mean age = 19) and older adults (n = 25, mean age = 69.5). We matched participant accuracy by titrating response window length. Both groups achieved similar performance and strategy during the titration, enabling a comparison without confounds resulting from differences in default age-specific strategies. All participants were then given monetary incentives to perform better in terms of accuracy. Both groups responded with enhanced accuracy, but the young participants improved much more, outperforming older adults, and reversing the speed-accuracy strategies that are typically observed. In addition, older participants reported higher baseline levels of motivation alongside a reduced motivation to alter performance for money. So, while the older participants could match young participant performance in titration due to their higher baseline motivational levels, the young participants improved much more than older adults in response to the monetary incentive. From these findings we argue that older adults are intrinsically motivated to do well on tasks whereas younger age groups perform optimally only after incentivisation.

## Introduction

### Age effects in sustained attention

The Sustained Attention to Response Task (SART) [1] has been widely used to study sustained attention in both clinical [2] and healthy populations [3]. It is mainly used as a short probe of failures in attention to reflect lapses in vigilance, but has been

**Data availability statement:** All experimental data files and the analysis and data collection scripts are available from the osf.io database (https://doi.org/10.17605/OSF.IO/XP7MK).

**Funding:** SH was supported by the Economic and Social Research Council Grant: ES/P000681/1. The funder had no role in the study design, data collection and analysis, decision to publish, or preparation of the manuscript. This does not alter our adherence to PLOS ONE policies on sharing data and materials.

**Competing interests:** The funders had no role in study design, data collection and analysis, decision to publish or preparation of the manuscript.

increasingly used to investigate diverse factors influencing attentional responses in the healthy population. This has led to the identification of age-specific behavioural patterns during SART performance [4,5]. Older participants typically show higher accuracy on nogo trials, or trials when response is withheld [6], and have thus been reported as prioritising accuracy in their response [4,7]. Conversely, their longer reaction times [4,8] are often understood to reflect the general decline in sustained attention ability arising from ageing [9] or a general decline in processing speed linked to changes in key brain areas involved in its maintenance [10]. Older participants may use a longer processing window to counteract this decline in processing (and instead maintain high accuracy [11]) so this older participant accuracy advantage has been framed as an adaptation to reductions in processing speed [12]. On the other hand, because this difference between young and older participants could simply reflect age-dependent strategic choices in task execution (a different argument put forward previously [9,13]), an interpretation of the observed performance differences in terms of an effect of ageing is questionable. In a recent study [14], although we replicated the age-dependent performance strategies (high accuracy and slow responses in older adults, low accuracy and high response speed in young adults), importantly we did not find vigilance decrements [15] that we expected to observe from time-on-task fatiguing mechanisms [16,17] in either age group. We therefore highlighted the need to identify different factors leading to age-dependent differences in overall vigilance (including strategy choices, motivation, resilience to fatigue), to better understand the general effects of ageing on sustained attention.

The default parameters of the SART [1] provide the participant with an ambiguous choice of prioritising either speed or accuracy. The participant is incentivised to decide their own strategic priority, based on unmonitored internal processes [18,19]. A dichotomy in strategic response is thus enabled by a sufficiently long response window in the default task design. Group-specific strategies then emerge [4,20] because participants choose different points in the window to respond: Young participants typically react early in the response window, displaying faster reaction times [6]. In contrast, older participants tend to more fully utilise the length of the window and in so doing, increase their accuracy [8,21].

Previous studies have used modified versions of the SART to investigate underlying mental processes that may influence performance [22–24] including manipulations of task complexity [25,26] to affect strategy choice. We follow this strand here by manipulating the speed-accuracy trade-off [27,28] in the strategy choice between accuracy and reaction time. We achieve this by imposing a varied response window length, eliciting faster response times by necessity and thus reducing participant accuracy. In titrating [29–31] to a pre-defined accuracy constant we aimed to unify the strategy across both age groups and thus reveal underlying differences in the performance of each group [32–36].

## Motivation

Researchers have already stated an effect of age on strategy choice as underpinned by differences in levels of baseline motivation. Definitions of motivation may vary,

but are commonly linked to reward [37]. Intrinsic motivation is generally characterised as an interest or enjoyment in the task stemming from the individual [38]. Multiple studies describe older adults as highly intrinsically motivated participants [14,39–41]. Motivation has previously been noted to underlie the surprising behavioural advantage in older adults [4], biasing them towards a more motivationally-demanding accuracy strategy [36]. In other related investigations, older adults were noted to opt for a more self-driven inhibitory strategy, again leading to the pattern of longer reaction times and higher accuracy [8,21]. Others have shown older adults to be less prone to shift their strategy in response to further motivators due to ceiling motivation levels arising from their values [40]. They are considered to experience higher rewarding value from the onset of the experiment, stemming from their beliefs of a benefit to society and a positive contribution in participation in research [39]. In addition, older adults have been shown to have less sensitivity to reward and punishment, limiting alterations to their strategy [42,43]. It is even possible that older adults may experience a paradoxical worsening reaction to reward initiatives [44].

This increased baseline level of motivation in older adults can be contrasted with the bias present in a young student sample. Samples exclusively relying on a population of psychology students were previously criticised for low internal validity [45]. The monetary reward used as a means of sampling participants for experiments was suggested to carry a confounding effect [46,47]. Specifically, student participants have been noted to rely on a strategy of conservation of effort [48], while also showing higher mind-wandering levels when compared to other samples [6]. In this experiment, after titration, we introduce a surprise (monetary) motivational intervention to test for any resulting performance divergence between the age groups.

### Fatigue

Fatigue is another factor considered to impede performance during sustained attention, with several studies reporting heightened levels of subjective fatigue accompanying time-on-task effects [15,49–53]. While some work has highlighted an effect of fatigue on behaviour, we failed to detect a reliable relationship with SART performance in our recent work [14], amongst other investigators who also failed to find a reliable link [54,55]. It has been theorised that motivational effects may contribute to the assumed behavioural effect of fatigue [56,57]. The present study will thus also include a measure of recently experienced fatigue to re-test its possible impact on behaviour.

### Study rationale

In previous research, the introduction of an objective reward as a motivational manipulation led to both an increase in speed and accuracy, yet so far this has been tested only in a young, student sample [27,28,37]. It thus remains unclear how different age groups perform in response to a motivational initiative once their underlying strategy is unified, or in fact whether they differ in response to a manipulation of motivation. The precise relationship of motivational changes to age-specific performance in sustained attention is thus addressed in the present experimental design: we first aligned young and older participants to the same (higher accuracy over reaction time) strategy by titration of the task difficulty and then introduced a surprise monetary incentive. We predicted that inherent lower motivation would elicit a stronger motivating effect of the surprise motivational intervention, leading to a greater accuracy in the motivational block. We also predicted young participants to have lower starting levels of motivation and that their accuracy improvement would be greater after the surprise motivational intervention than that of the older age group.

## Methods

### Participants

The experimental design and hypotheses were pre-registered on the Open Science Framework (https://osf.io/pyzn7). The study was approved by the University of Glasgow College of Medical and Veterinary Life Sciences Ethics committee

(Approval number: 200230387). All participant data was acquired between the dates 10th of October 2024 and 21st of November 2024. A total of 56 healthy adults were recruited between the ages of 18 and 96 from the university subject pool and local area and given monetary compensation for their time. Written consent was acquired from all participants. Participants were balanced for gender and were asked to report any existing medical conditions, eye-sight correction and medications which might impact their performance. Six participants were excluded throughout data collection: One participant reported an uncorrected visual deficiency in the left eye, as also detected by a visual field test. One participant was excluded for excessive caffeine use (2 units above recommended dosage). A further participant was excluded for reporting poor sleep (4 hours per day). Two participants were excluded for low MoCA scores (<24). Finally, a participant was excluded due to a possible technical fault, or inaccurate attendance to instructions (go accuracy lower than 80% throughout multiple blocks).

The final sample consisted of 50 participants (F = 28, M = 21, NB = 1) based on a power analysis of the sample needed to acquire an effect size of f = 0.2 in a 2x2 ANOVA within-between factor interaction. The participants were divided into a young (M = 25, F = 16, NB = 1, mean age = 19, SD = 1.38, range = 18–23) and older (n = 25, F = 13, mean age = 69.5, SD = 6.72, range = 60–85) age group. Five participants were left-handed, one was a smoker, all reported low to moderate caffeine consumption (estimated mean units per day = 1.09, SD = 1.05, range = 0–4), matching the maximum recommended daily dose of 400 mg of caffeine [58]. They also reported an average of 7.2 hours of sleep per day (SD = 1.04, range = 6–12). All young participants were enrolled university students. The participants were screened for cognitive difficulties using the Montreal Cognitive Assessment test (MoCA; [59]), reflecting scores representative of a healthy population [60] in both young (mean score = 29.2, SD = 1.7, range = 25–31) and older adults (mean = 27.7, SD = 1.65, range = 24–30). A Welch's t-test showed lower MoCA scores in the older group, t(48) = 3.12, p = .003, d = .883, as expected when comparing young and older populations [61]. Cut-offs for the age groups were defined as 2 SDs below the mean [60], meaning all participants with a MoCA score below 24 were excluded. A short (4-minute) computerised visual screening assessment was administered at the beginning of the session to exclude potential visual pathologies. The task was adapted from a previous experiment on young and older groups [14]. A Welch's t-test identified no age-group differences in target detection within the visual regions where the SART stimuli were to be presented, t(48) = 1.57, p = .128, d = .500.

## Procedure

The experimental task and procedure are outlined in Fig 1. Participants provided basic demographic information and self-reported any known impediments to participation. After this, they first completed a measure of trait fatigue (MFI) and a

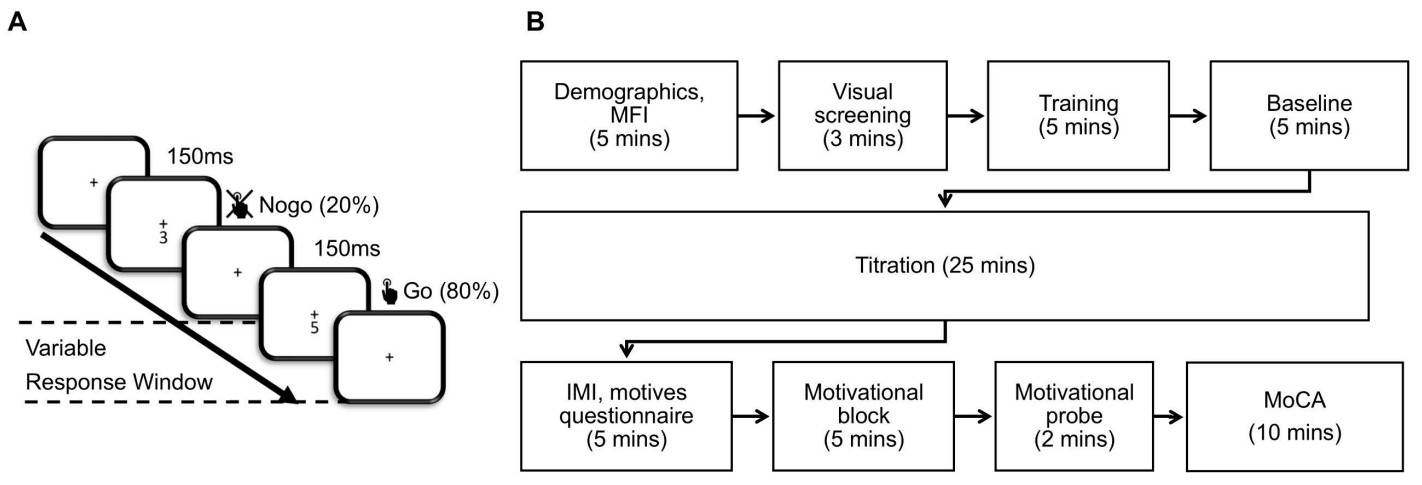

**Fig 1. Experimental procedure.**

brief adapted visual screening test [14] targeting their central visual area to detect any impairments preventing them from participation. Participants proceeded to a brief training session to familiarise them with the SART. If they were unable to achieve the minimum required standard in the experiment during 2 mins of non-titrated SART (above chance accuracy), the training session was repeated. They then undertook one 5-minute baseline block of the SART matching our previous experiment [14]. The participants then carried out an adapted version of the task, designed for a titration-based investigation of performance. They were instructed to be as accurate as possible, but to respond before the onset of the next trial. The difficulty levels of the task were manipulated through either an increase or decrease of the response window length by 50ms, based on the participant's accuracy in sets of 25 trials and determined by a target criterion of 92% accuracy. In total, participants carried out the titration procedure for 25 minutes, divided into 4 blocks with short breaks in between. They were not informed how many blocks of the task they were expected to complete in total. This procedure aimed to determine the response window length (300ms – 2500ms) at which the participant consistently achieved an overall accuracy close to 92%.

At the end of the titration, the participants were informed that the experiment was finished, and completed a subjective intrinsic motivation inventory [IMI; 62] and a brief adapted subjective measure to record specific motivations behind their experimentation. They then took a self-paced break. Then, all participants were informed about a further, unanticipated, block of the task presented at the same difficulty level as the last. They were likewise informed that if they achieved the highest improvement in accuracy, relative to all other participants in their age group, they would receive a prize of £50. The participants then carried out a final 5-minute block of the task, set to the difficulty level matching the average response window length of the last 125 trials at the end of the titration period, but with no further titration. At the end of the motivational block, they proceeded to fill in a single item measure (VAS-M) on perceived changes in their motivation as a result of the initiative. Before the end, they were screened for any cognitive impairments that could impact the experiment using the MoCA [59] and then proceed to be debriefed. The overall duration of the experiment was 65 minutes.

## Task

The participants underwent a modified version of the SART with varied levels of difficulty, implemented in PsychoPy, using custom Python scripts [63]. The task was displayed on a digital monitor (Dell Optiplex 9010), with a screen resolution of 1280x1024 pixels and a refresh rate of 60 Hz. Participants were seated 60 cm from the screen, maintaining horizontal eye level with the centre of the display by the use of a chin rest. In each trial, the participants were instructed to fixate centrally on the fixation cross and attend to a stimulus presented at an angular distance of 1°, consisting of a number between 0–9 presented centrally for 150ms. The number then disappeared during the response window, which had a variable duration of 300–2500ms, before the next trial started. The response window length and the learning block response window lengths were set to 1000ms at the start of the experiment for all participants. The task was to respond using a spacebar press to all numbers that appeared (go trials), apart from the numbers 3 or 6, whilst withholding response to the appearance of numbers 3 and 6 (nogo trials). The numbers were pre-generated to be distributed randomly and represented in equal frequency. Based on the accuracy of the participant in a set of 25 trials, the subsequent set of trials had their response window length shortened or lengthened by 50ms to eventually achieve a desired equilibrium (92% accuracy for each participant). If accuracy on the previous block was lower than 92%, the subsequent block was made easier by lengthening the response window by 50ms. If accuracy was exactly 92%, the response window was kept constant. An accuracy of 92% was chosen to correspond to 60% nogo accuracy and 100% go accuracy (corresponding to 20% nogo trial rate), since go trial accuracy was expected to be at ceiling level for most participants [5]. An additional static fixation period of 6s was added between the sets of 25 trials. The difficulty of the motivational block was calculated to represent the average response window length, rounded to the nearest increment of 50ms in the last 125 trials of the titration, to reduce the effects of random fluctuations in accuracy.

## Measures

The participants were asked to report their age, gender, number of hours of sleep in the past week and caffeine intake on the day as well as disclose known impediments to participation.

The Intrinsic Motivation Inventory (IMI) was used as a measure of subjective motivation [62]. It has been recently used for valuation of motivation and cognitive task performance [64] and continues to show good reliability (Cronbach alpha > .7, [65]). It is a 7-point Likert scale that contains 45 items spread across 7 subscales. The three most relevant subscales were used: interest (7 items; e.g., 'I enjoyed doing this activity'), effort (5 items; e.g., 'I put a lot of effort into this') and value (7 items; e.g., 'I think this was an important activity') subscales. The experiment further used a motivation item question adapted from the use in our lab, probing participants for reasons for taking part in the experiment (8 options; e.g., 'To help the researchers make new scientific discoveries'). The participants were also probed on a visual analogue scale for motivation (VAS-M) with values 0–100 and a single question on the extent they felt motivated by the motivational intervention.

The Multidimensional Fatigue Inventory (MFI) [66] was used to measure trait fatigue, and was comprised of 5 subscales with 4 items each (20 items in total) on a 5-point Likert scale. Previous work indicated a very good reliability of $\alpha = .84$ and showed a lack of floor and ceiling effects as well as item redundancy [67].

## Results

All analyses were carried out in R (R Core Team, 2024) using the packages 'tidyverse' [68], 'psych' [69], 'psycho' [70], 'BayesFactor' [71], 'lsr' [72] and 'ez' [73]. Packages used for graphical depiction were: 'ggpubr' [74], 'viridis' [75] and 'Cairo' [76]. Any trials with a reaction time < 150ms were excluded from the analysis as likely to be representative of anticipation error [77] (.6%). In addition to the pre-registered aims, trials with reaction times of excessive length defined as 3 standard deviations above the age group mean were also excluded (1.29%, see Vankov for a discussion of acceptable trimming methods [78]). Reaction times further showed a skew (0.835), and so were log-transformed for any subsequent analysis. While not originally included in the pre-registration, D-prime ($d'$) was chosen as a useful addition to further measure participant sensitivity in the context of possible changes of task strategy [79]. $D'$ was computed using the 'psycho' package [69], using the function *dprime*, following the formula in Stanislaw and Todorov [80] and applying an adjustment for extreme values from Hautus [81]. Correct go trials were considered hits, incorrect go trials misses, correct nogo trials correct rejections and incorrect nogo trials false alarms. As per previous findings [79], the achieved mean $d'$ values fell in the range expected for the SART.

Since each participant completed the task for a fixed duration of 25 minutes, the total trial numbers differed among the participants due to the variable response window lengths. A t-test on the total number of titration period trials in young (mean = 954.89, SD = 138.52, range = 700−1150) and older adults (mean = 99.25, SD = 125.41, range = 700−1275) showed no differences between the groups, t(48) = −1.14, p = 258, d = .305. A t-test was also run on the average window length between the young (mean = .880, SD = .217, range = .647-1.41) and older adults (mean = .827, SD = .190, range = .523–1.38) in the whole titration period, also showing no between-group differences, t(48) =.919, p = .363, d = .260.

As the frequentist approach taken in our analyses resulted in multiple borderline significant effects (see below), we added a Bayesian analysis follow-up approach (following Keysers et al. [82]) approach to help quantifying the strength of evidence for either the alternative ($H_1$) or the null hypothesis ($H_0$). Bayes factors (BFs) were reported to express the likelihood of the data under $H_1$ relative to $H_0$ (BF < 1/3 indicating support for $H_0$, and BF > 3 indicating strong support for $H_1$, and values between 1/3 and 3 suggesting that the data are insensitive or only provide anecdotal evidence for the respective hypotheses [83]). All t-test and ANOVA Bayesian equivalent tests were conducted using the 'BayesFactor' R Package [71]. Informed by our earlier work [5,14], the prior scale value was pre-set to 1/2 through rscaleFixed = "medium" for t-test and and rscaleRandom = "medium" for ANOVAs, respectively.

## Age-specific strategies

We first investigated differences among the age groups in the baseline block. A between groups t-test showed no differences between the two age groups on nogo accuracy in the baseline SART block, t(48) = −.200, p = .421, d = .057, depicted in Fig 2A. A between-groups t-test showed that the older group had higher reaction times in the baseline SART block, t(48) = −1.77, p = .042, d = .500, as reflected in Fig 2B. As this constituted a marginally null finding prior to the removal of the long reaction times, a Bayes factor equivalent was run, showing no evidence for either of the hypotheses ($B_{10}$ = 1.00, ±.010%, $H_1$: age difference in reaction times). In light of the marginal significance and the bayes test, we interpret this finding as underpowered, yet hinting at the expected age effect, with higher reaction times in older adults.

Additionally, to test age effects after titration, a two-sample t-test was run, testing the difference between young and older participants on experimental difficulty level (combined trial and presentation length) at the titrated window length. This again showed no differences between the groups, t(48) =.608, p = .273, d = .172, as depicted in Fig 2C.

The findings thus indicate that we did not replicate age-specific strategies in our sample, but that both groups had similar performance levels throughout the period prior to the motivational monetary manipulation.

## Titration

Although not pre-registered, we tested the impact of the titration procedure on participant performance. The goal of this analysis was to demonstrate (as a sanity check) that the titration did indeed work in terms of performance changes in all (high and lower performing) participants, via the adapting of the response window. We did this across the whole sample as an age split would have underpowered this essential validation of our titration method.

Participants were split based on their initial median accuracy (93.50%). This resulted in two groups: those who had better accuracy than the average at baseline (mean = 95.80%, SD = 1.03%) and those who were worse in accuracy than the average (mean = 91.0%, SD = 2.91%). Then, the performance of both groups across the four titration blocks was modelled on several behavioural metrics.

A multiple linear regression [F(3, 196) = 16.74, $R^2$ = .19, p < .001, $f^2$ = .237] showed that better performers at baseline had higher accuracy over all 4 blocks: β = −.052, t = −5.81, p < .001. A main effect of titration block showed that overall accuracy decreased throughout the experiment, β = −.013, t = −5.65, p < .001. There was an interaction between the effects

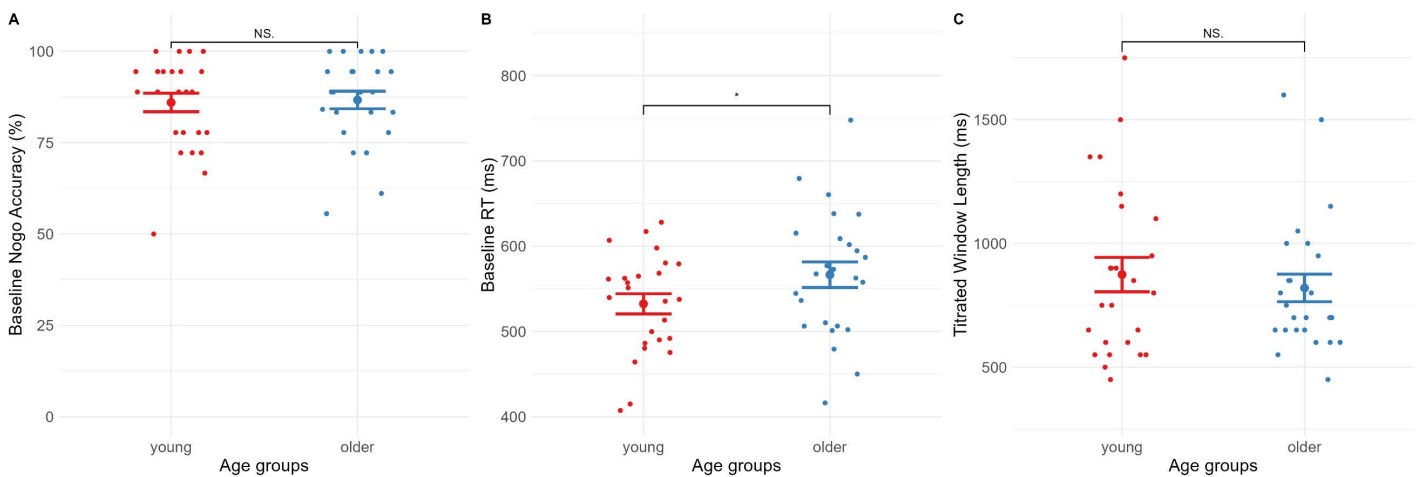

**Fig 2. Age differences at baseline and after titration.** Young participants did not differ from older participants in the baseline comparison block (5 minutes) neither in nogo accuracy **(A)** nor reaction times **(B)**. Additionally, age groups (young and older) did not differ in their titrated window lengths at the end of titration **(C)**.

of group and titration, β = −.015, t = 4.48, p < .001. A series of post-hoc t-tests was conducted to outline the titration blocks where the groups differed. A t-test found a difference between the groups in block one, t(48) = 7.72, p = .001, but not in block two, t(48) = 1.33, p = .19, block three, t(48) = −.576, p = .57 or block four, t(48) = .875, p = .386. The groups were thus no longer different in their overall accuracy by block two. Collectively, this indicates that the gap between the performers in overall accuracy decreased over time and disappeared, as seen in Fig 3A.

A multiple linear regression [F(3, 196) = 19.74, $R^2$ = .22, p < .001, $f^2$ = .283] showed that better performers had higher nogo accuracy β = −.189, t = −5.16, p < .001. A main effect of titration block showed that nogo accuracy decreased throughout the experiment, β = −.034, t = −3.69, p < .001. There was an interaction between the effects of group and titration, β = .035, t = 2.64, p = .009. A series of post-hoc t-tests was conducted to test in which titration blocks the groups differed. A t-test found a difference between the groups in block one, t(48) = 7.44, p < .001 and block two, t(48) = 2.70, p = .01. No difference was found in block three, t(48) = 1.35, p = .185 and block four, t(48) = 1.51, p = .176. Collectively, this indicates that the gap between the performers in nogo accuracy decreased over time and disappeared, as seen in Fig 3B.

A multiple linear regression [F(3, 196) = 7.73, $R^2$ = .092, p < .001, $f^2$ = .101] showed that better performers had higher reaction times, β = −.113, t = −2.72, p = .007. A main effect of titration block showed that reaction times generally decreased throughout the experiment, β = −.025, t = −2.40, p = .017. There was no interaction between the effects of group and titration, β = .017, t = 1.14, p = .256. As this previously constituted a marginally null finding (prior to the removal of the long reaction times), a bayes factor test was run, showing highest evidence in favour of the model *Perfomance group + Block* without an interaction, ($B_{10} > 5, \pm 3.43\%$). This thus further suggests an unlikely interaction between block and age groups.

Collectively, this indicates that better performers preserved their slower reaction times throughout the titration period relative to worse performers, but both groups generally reduced their response times, as seen in Fig 3C.

A multiple linear regression [F(3, 196) = 32.89, $R^2$ = .325, p < .001, $f^2$ = .481] showed no difference between the groups in their response window length: β = .072, t = 1.12, p = .263. A main effect of titration block showed that response window length decreased throughout the experiment, β = −.059, t = −3.65, p = .001. There was an interaction between the effects of group and titration, β = .068, t = 2.90, p = .004. The interaction indicates that better performers gradually achieved more narrow (hence difficult) response window lengths, with high performers reaching a relatively low response window length and low performers retaining a high response window length, as seen in Fig 3D.

Collectively, the testing of performance over time confirms that both groups of performers achieved an average of 92% accuracy at the end of the titration blocks. Good performers, in addition, reached a shorter response window length while matching the same accuracy level. The titration thus generally raised the difficulty of the task for high performers and maintained or reduced the difficulty for poorer performers. Alongside this, there was a limited effect on reaction times, with high performers preserving higher reaction times to maintain a more accurate strategy.

## Motivational manipulation

The following analysis investigated the effects of the surprise motivational intervention. A 2x2 mixed ANOVA between age groups (young, older) and time points (last 125 trials of titration, whole motivational block) was run on overall accuracy. The resulting model showed no main effect of age, F(1, 48) = 2.50, p = .120, η² = .029, but that all participants were more accurate after the motivation, F(1, 48) = 48.27, p < .001, η² = .302. A significant interaction, F(1, 48) = 4.37, p = .042, η² = .038, showed that the young participants improved much more than the older adults, as depicted in Fig 4A. A Bayes Factor test was run to reconstruct the 2x2 ANOVA. Of all possible resulting models (n = 7), the model with the highest Bayes Factor was *Block + Age group:Block*, providing strong evidence for $H_1$ (BF > 5, ±.790%). Thus, it confirmed that the motivational manipulation worked to increase accuracy in both groups, but more so in the young group.

A 2x2 mixed ANOVA between age groups (young, older) and time points (last 125 trials of titration, whole motivational block) was run only on nogo accuracy, showing the same pattern. There was no main effect of age, F(1, 48) = .001, p = .974, η² < .001, but the participants were more accurate after the intervention, F(1, 48) = 39.33, p < .001, η² = .163. A

 

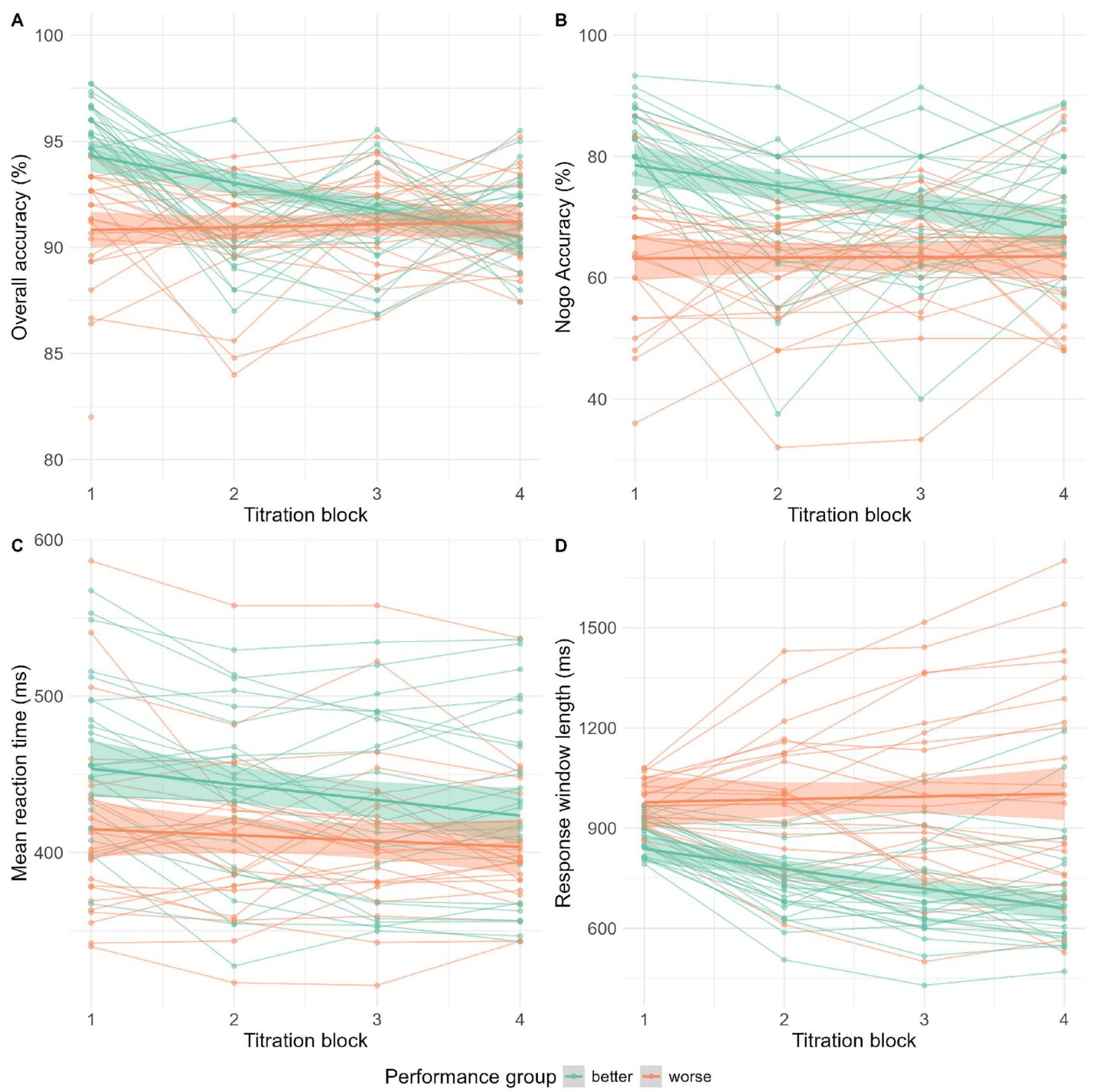

**Fig 3. Median-split participant performance in titration over time.** Participants were split into better (green) and worse (orange) performance groups based on their overall accuracy in titration block 1. The performance of each of these groups was then plotted across all four titration blocks in the metrics of overall accuracy **(A)**, nogo accuracy **(B)**, mean reaction time (C) and titrated response window length **(D)**.

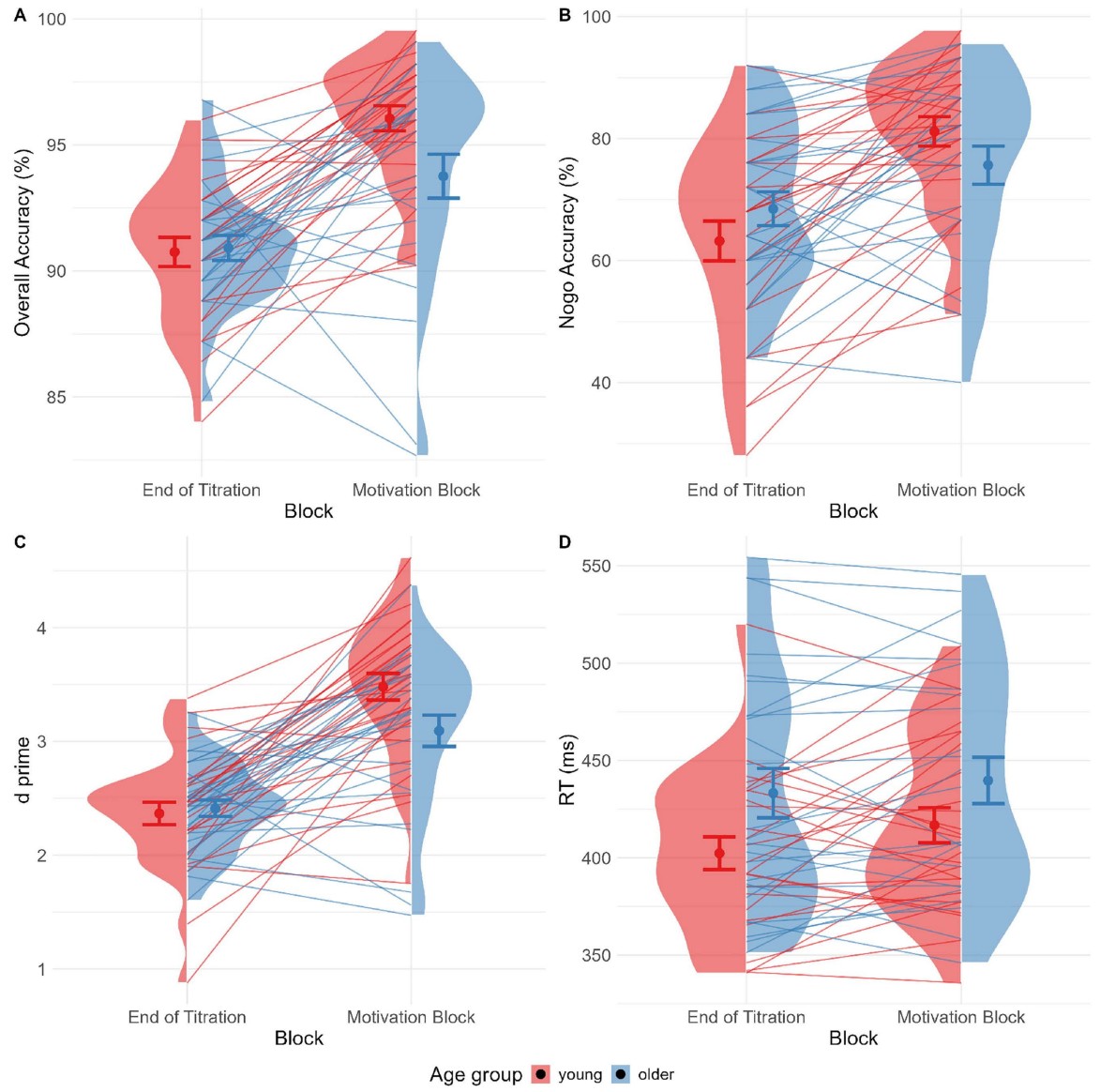

**Fig 4. Performance difference after motivational manipulation.** Plots comparing participant performance at the end of the titration period (last 125 trials of titrated block 4) with performance during the Motivational block. Young participants improved significantly more than older participants after the motivational manipulation both when measured in overall accuracy **(A)**, when focusing on nogo accuracy **(B)** or sensitivity captured by d' **(C)**. No clear differences were present across reaction times **(D)**.

significant interaction, $F(1, 48) = 7.26$, $p < .001$, $\eta^2 = .035$, showed that the young participants improved much more than older adults, as seen in Fig 4B. As per previous analysis, a Bayes Factor equivalent to a 2x2 ANOVA was run, testing the effect of age group and time point on nogo accuracy. Across all possible models (n = 7), the model showing highest evidence in favour of $H_1$ was *Time Point + Age Group:Time Point* ($B_{10} > 5, \pm .680\%$), confirming that the motivational manipulation worked to increase nogo accuracy in both groups, but more so in the young group.

In addition, we were interested in testing the same effects on *d'* as the chosen measure of sensitivity. A 2x2 mixed ANOVA between age groups (young, older) and time points (last 125 trials of titration, whole motivational block) was

run on *d'*, showing the same pattern as the previous tests of accuracy. There was no main effect of age, $F_{(1, 48)} = 2.07$, $p = .156$, $\eta^2 = .022$, but the participants were more accurate after the intervention, $F_{(1, 48)} = 81.80$, $p < .001$, $\eta^2 = .383$. A significant interaction, $F_{(1, 48)} = 4.75$, $p = .034$, $\eta^2 = .035$, further showed that the motivational manipulation worked to increase sensitivity in both age groups, but that young participants"sensitivity improved much more than that of the older adults, as seen in Fig 4C.

A 2x2 mixed ANOVA between age groups (young, older) and time points (last 125 trials of titration, whole motivational block) was run for reaction times. There was no main effect of age, $F_{(1, 48)} = 3.40$, $p = .072$, $\eta^2 = .060$, but the participants were slower after the motivational manipulation $F_{(1, 48)} = 8.36$, $p = .006$, $\eta^2 = .019$, with no significant interaction, $F_{(1, 48)} = .716$, $p = .402$, $\eta^2 = .002$, indicating a similar pattern of slowing in both age groups. The subsequent Bayes Factor test reconstructing the 2x2 ANOVA deviated from this finding though. Of all the resulting models, the model including *Age group* had the highest Bayes factor, showing anecdotal evidence supporting the $H_1$ ($B_{10} = 2.12, \pm 0.01\%$). Because of the inconsistency across the frequentist and Bayesian results, we draw no conclusions here, despite the overall differences in reaction times slowing, as per Fig 4D.

## Motivational differences

Next, we investigated the differences among age groups in the subjective perception of their motivation.

A Cronbach's alpha was calculated for each of the subjective scales. IMI – interest showed alpha = .841, IMI – effort alpha = .765, IMI – value alpha = .821, indicating good to excellent reliability of the measures.

A between groups t-test was run, testing for differences between young and older participants on intrinsic motivation upon completion of the titration, for each of the three motivation sub-scales. Older participants had higher subjective motivation on IMI – interest, $t_{(48)} = -2.23$, $p = .015$, $d = .632$ (Fig 5A), and IMI – value, $t_{(48)} = -2.31$, $p = .013$, $d = .655$ (Fig 5B), but not on IMI – effort, $t_{(48)} = -.038$, $p = .485$, $d = .011$ (Fig 5C).

A t-test was run between the two age groups on post-motivational block change in motivation measured by the visual analogue scale (VAS-M). The young group was significantly more motivated by the monetary incentive compared to the older adults, $t_{(48)} = 6.40$, $p < .001$, $d = 1.81$, seen in Fig 5D.

A Pearson's chi-square test assessed the difference between the young and older participants in the distribution of their reported reasons for taking part in the experiment. The test did not show any differences between age groups, $X_{(36)} = 42$, $p = .227$. Arguably, the findings were underpowered to adequately detect differences among the two age groups as out of the 8 reasons for participation, some cell observations in reasons for participation fell under 5 [84]. Nevertheless, the young participants were informatively much more motivated by money to participate (23 young vs 5 older).

## Connection to fatigue

We also explored whether levels of motivation and subjective trait fatigue were associated with performance on the SART.

Cronbach's alpha was calculated for each of the subjective subscales of the multidimensional fatigue inventory. Most of the scales had good reliability: MFI – general fatigue had an alpha = .82, MFI – mental fatigue alpha = .843, MFI – physical fatigue alpha = .813, MFI – reduced activity alpha = .757, but MFI – reduced motivation only showed low alpha = .578.

Multiple linear regressions were run between the two age groups on subjective fatigue scores and titrated window length in the titration SART block, one for each subscale of MFI. No prediction of titrated window length or age group by fatigue was found for MFI scores overall, $[F_{(3, 46)} = .676, R^2 = 0.020, p = .571]$, MFI general fatigue $[F_{(3, 46)} = .399, R^2 = 0.038, p = .754]$, MFI physical fatigue $[F_{(3, 46)} = .453, R^2 = 0.035, p = .716]$, MFI mental fatigue $[F_{(3, 46)} = 1.018, R^2 = 0.001, p = .393]$, MFI reduced activity $[F_{(3, 46)} = .670, R^2 = 0.021, p = .575]$ or MFI reduced motivation $[F_{(3, 46)} = .592, R^2 = 0.026, p = .623]$.

A multiple linear regression tested the difference between the two age groups on the correlation between total subjective fatigue scores and total intrinsic motivation scores $[F_{(3, 46)} = 6.08, R^2 = .237, p = .001, f^2 = .311]$. Older

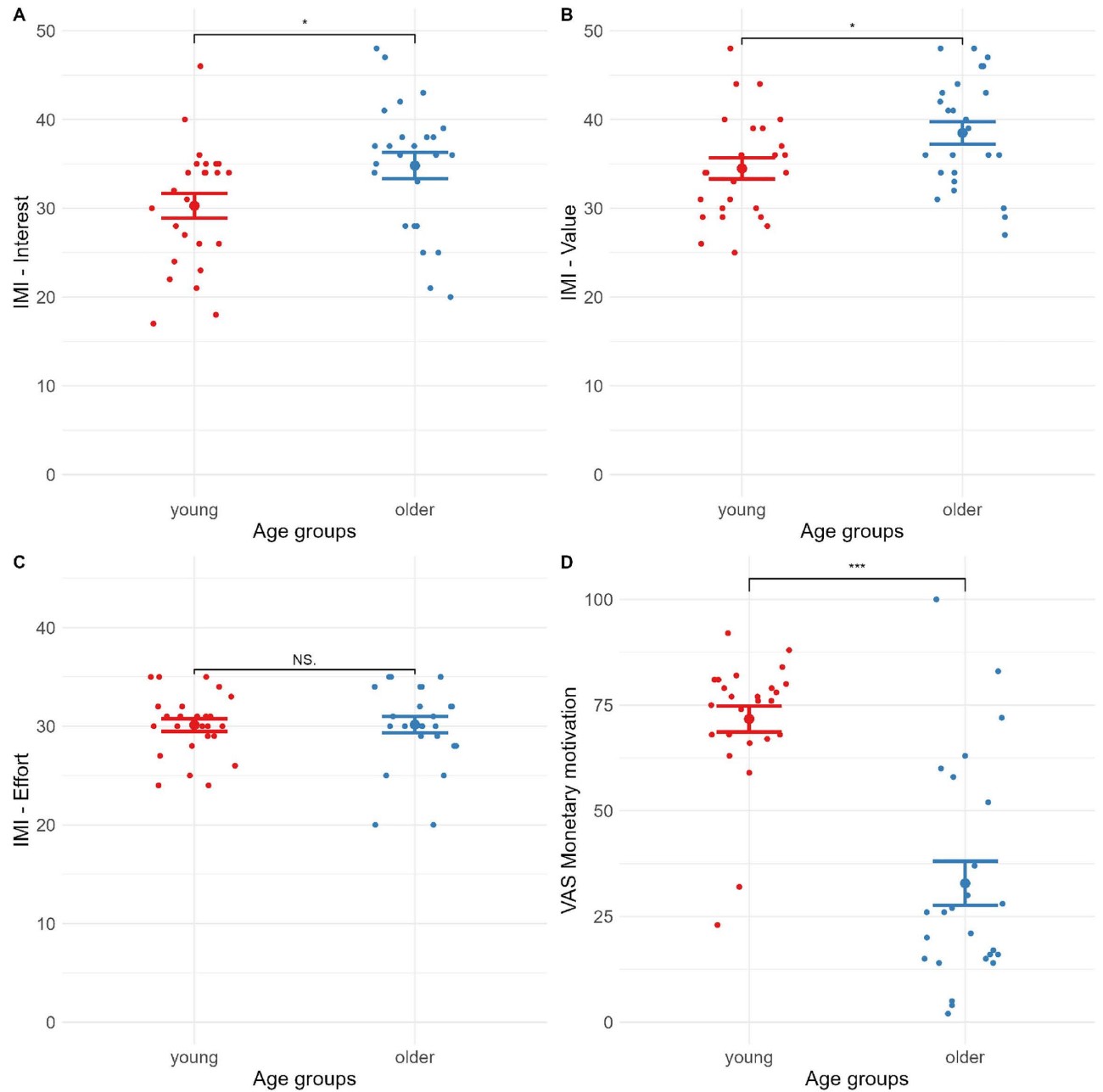

**Fig 5. Age differences in motivation.** Older participants had higher motivation to undertake the task based on IMI – Interest **(A)** and IMI – value **(B)**, but not IMI – effort **(C)**. The older participants were also less motivated by the intervention than younger participants **(D)**.

adults were more motivated than young adults, β = 47.43, t = 3.09, p = .003, with no main effect of fatigue, β = .164, t = .669, p = .507 and with a significant interaction between age group and MFI total fatigue, β = −.906, t = −2.78, p = .008. An inspection of a Fig 6 depicting the relationship shows that there was no relationship between motivation and fatigue in young participants, but older participants experienced more motivation if they also experienced being less fatigued.

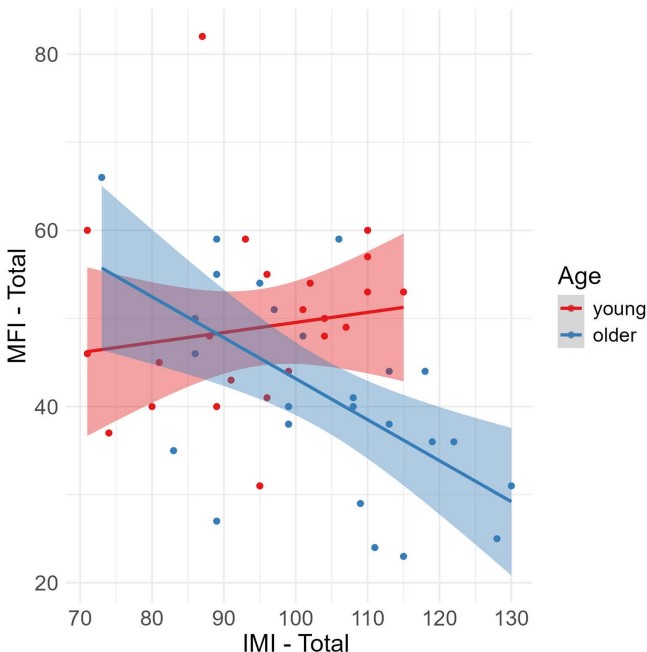

**Fig 6. Age differences in the relationship of fatigue to motivation.** Older participant total scores on the Multidimensional Fatigue Inventory (MFI) were associated with lower total scores on the Intrinsic Motivation Inventory (IMI), with no such relationship in young participants.

## Discussion

This study examined how age, motivation, and fatigue influence strategy choice in the SART. To ensure comparable performance of both age groups, the response window was individually adjusted by titration to attain a shared accuracy of 92%. Notably, older participants reached the target accuracy without, as a group, requiring longer (and thus easier) response windows than the young adults. As a result, we did not replicate the prominent age effect of older adults being accurate but slower, and young adults being fast but inaccurate [4,5]. Nevertheless, the titration procedure still ensured that accuracy was matched before all participants were incentivised to perform better. This manipulation then success-fully elicited an age effect: older adults showed less improvement than younger participants, who became much more accurate.

Both age groups initially reached a comparable level of task difficulty, as indicated by the similar response window lengths at the end of titration. During this phase, older adults also reported higher levels of subjective motivation, aligning with previous findings about their higher intrinsic motivation [85]. We thus propose that their intrinsic higher motivation enabled this group of older adults to keep pace with the younger group up to the point of the motivational manipulation [86,87]. Younger adults in turn were less motivated to do the task, which likely explains their (relatively) poorer perfor-mance prior to incentivisation and their greater ability to subsequently improve. In a related study, DeRight and Jorgensen [88] investigated low effort in a sample of college students and also found a proportion of students with low effort, resulting in a surprising, subthreshold performance on key attentional and cognitive tasks. Dunn and colleagues [89] reached a similar conclusion with a motivational imbalance in the student sample. The young participants in our study may be seen as exerting the minimum effort required to meet task demands, thereby adopting a strategy aimed at conserving energy [44,90,91] until there comes a point of renewed interest; in this case the interest is renewed by an extra monetary reward [92]. This parsimonious strategy is arguably advantageous in light of the relatively low perceived value of the experiment,

which then changes as a result of the motivation manipulation: our young participants strategically limited their effort during the titration and then improved heavily once motivated.

We thus highlight, for the first time in sustained attention research, motivation as the primary factor driving age-related differences in performance. The young participants reported strong reactivity to the motivational manipulation and showed greater improvement in accuracy compared to the older adults. The older sample did not alter their performance as much after the motivation manipulation and reported a low perceived effect of the monetary initiative.

In the existing literature, diverse reasons are in discussion regarding the increased presence of intrinsic motivation in the older participants. It has been suggested that older adults seek to compensate for age-related slowing and so become more motivated to perform well [8,13]. This is then translated into a more demanding behavioural strategy. The account is based on reports of strong rapport between older adults and the researcher [40,93] and a strong self-reported positive valuation of experimental participation associated with older adults [94]. According to the socioemotional selectivity theory [40,41,95], older participants experience the positive value of participation as they are more attracted to short-term goals directly related to their experience during the experiment. They maximise the emotional well-being experienced in the present moment, gained from volunteering in scientific research [96]. Young participants conversely perceive time as a vast resource and so prioritise future-oriented goals, including advancing their future socio-economic status, and thus responding to monetary rewards. As a result, older adults may perceive participating in research itself as more rewarding than the financial compensation. This could further explain why the motivational manipulation only showed a limited improvement in older adults, who reported that the prospect of a future reward did not impact their already strong investment in the experiment. Their limited responsivity likewise aligns with a report of a smaller inclination of older adults to switch to a different behavioural strategy, contrasting with the ability of young people to more readily change task strategy [97].

None of these effects were connected to fatigue and there were no differences in trait fatigue between the groups, aside from a link between fatigue and motivation in the older group. We speculate that this could indicate fatigue as a component of motivation in the older group, but more research is needed to test this connection directly. Given this minimal link to fatigue, we propose that the motivational differences at baseline and in response to additional monetary manipulation underlie the observed age effect in sustained attention.

## Limitations

It should be acknowledged that the study's recruitment approach was selective, thus making it potentially susceptible to bias in the form of higher socioeconomic status, health and educational levels relative to the typically ageing population. This was reflected in the matched high educational attainment and generally minor differences only in MoCA scores between the age groups [61]. Nevertheless, the conclusions regarding the older sample are aimed at populations typically participating in research. Older research participants may also differ from the typical population in their ability to access university-based research, and their interest in and awareness of opportunities to participate [98]. Future research may still consider ways to widen engagement, for example recruiting during public engagement events [99] may further aid the generalisability of the present findings. Our findings may also be amenable to replication in an online context (see our previous findings about strong age effects on sustained attention [5]).

## Future research

Our study raises the wider issue of the confounding effect of motivational factors in student samples commonly employed in experiments measuring performance. The present findings indicate that the choice of response strategy in young participants is dynamically affected by their level of motivation. Future studies, particularly those investigating the capacity of participants to perform at a certain level, should track motivational confounds. We thus show that on the SART, older participants were inherently motivated to do well, with only a little accuracy gain after the monetary incentive. In the case of this study, the commonly used young participant population was shown to generate suboptimal performance up to the

point of the extra motivational manipulation. We thus propose that older adults can be seen as intrinsically motivated to do well on tasks, whereas younger age groups perform optimally only after incentivisation. The approach of factoring subjective motivation into the study design may be further utilised and expanded by the use of more precise measurements of motivation, including follow-up probes of participation motives [100] or frequent probes during the task [49].

This experiment indirectly enriches the discussion of the theoretical underpinnings of vigilance. Poor overall vigilance in sustained attention has previously been suggested to correspond to fatigue. This was seen when fatigued populations performed worse in tasks requiring vigilance [17,101]. As further evidence, a vigilance decline would sometimes be observed during time-on-task in tiring tasks [49,102], yet this was not always the case [14,103–107]. The present findings indicate that the possible reason for this inconsistency may stem from a stronger influence of the motivation state on vigilance, more so than that of fatigue.

Finally, this study showed that the effect of motivation can be studied by targeting different participant age groups. As age has been previously strongly associated with the difference in performance during sustained attention [4,5], this experiment instead supports a motivational account of sustained attention performance differences [37,108]. It implies that overall vigilance can be better understood by incorporating the factor of motivation [109–111]. It also contributes to a possible explanation of the mixed efficacy of attempts to improve performance in students [112], pointing to the role of intrinsic motivation as an explanation.

## Conclusion

This study investigated the impact of motivation on age differences in performance during sustained attention. We showed that young participants' performance in sustained attention was improved by interference with their motivation levels much more than in a sample of older adults. Older participants reported higher baseline levels of motivation alongside a reduced motivation to alter performance for money. So, while the older participants could match young participant performance in titration due to their higher baseline motivational levels, the young participants improved much more than older adults in response to the monetary incentive. From these findings, we argue that older adults are intrinsically motivated to do well on tasks whereas younger age groups perform optimally only after incentivisation. The findings show the need to track motivational factors in investigations into sustained attention and likely apply to most studies comparing older and young samples.

## Author contributions

**Conceptualization:** Simon Hanzal, Gemma Learmonth, Gregor Thut, Monika Harvey.

**Formal analysis:** Simon Hanzal.

**Investigation:** Simon Hanzal.

**Methodology:** Simon Hanzal, Gemma Learmonth.

**Software:** Simon Hanzal.

**Visualization:** Simon Hanzal.

**Writing – original draft:** Simon Hanzal.

**Writing – review & editing:** Simon Hanzal, Gemma Learmonth, Gregor Thut, Monika Harvey.

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
