## [Decision Letter · Decision Letter 0]

5 Aug 2025

Dear Dr. Hanzal,

Thank you for submitting your manuscript to PLOS ONE. After careful consideration, we feel that it has merit but does not fully meet PLOS ONE’s publication criteria as it currently stands. Therefore, we invite you to submit a revised version of the manuscript that addresses the points raised during the review process.

We look forward to receiving your revised manuscript.

Kind regards,

David K Sewell

Academic Editor

PLOS ONE

Journal Requirements:

SH was supported by the Economic and Social Research Council Grant: ES/P000681/1.

The funders had no role in study design, data collection and analysis, decision to publish or preparation of the manuscript.

Additional Editor Comments:

Dear Dr Hanzal,

Thank you for submitting your manuscript to PLOS ONE. I have now received reviews from two experts, who are split in their evaluations. Reviewer 1 recommends rejection on the basis of the study achieving only limited insights into age differences in sustained attention. Reviewer 2 is more positive in their evaluation and recommends accepting the manuscript. My own independent reading of the manuscript falls closer to that of Reviewer 1 and so I am rejecting the manuscript.

My overall impression is that the work makes an interesting contribution and the use of the titration method is a very useful way of establishing functional equivalency in task performance across older and younger groups. However, like Reviewer 1, I was unclear about the rationale for the median split analyses (i.e., data visualized in Figure 3). Also, I think there are a few places where the relatively small sample size creates some issues in the data, power analysis notwithstanding. For example, on page 14, an interaction effect is dismissed with p = .051. I would be reluctant to completely rule out the possibility that the initially better performers sped up whereas the initially worse performers maintained a more consistent response speed over the duration of the task. In any case, there is a question of whether some of the null results are true nulls or if they reflect low power (as noted by Reviewer 1). Perhaps calculating Bayes factors to quantify relative support for the presence/absence of an effect would be a more illuminating way to go. Another option would be to opt for additional data collection, though there may be good reasons why this is not a viable option.

The biggest issue that leapt out to me, however, is the interpretation of the key data in Figure 4. In particular, the interaction showing a larger improvement in nogo accuracy in the younger group after the monetary motivation is introduced. I worry that the conclusions about much higher levels of improvement in younger participants are overstated, as (1) the younger group appears to suffer slightly poorer titrated performance and (2) it is not clear to me that incentivized performance actually differs across age groups. Admittedly, this criticism is tempered by the data showing overall accuracy, but the fact that nogo trials are much more diagnostic, I think a more detailed justification for your interpretation is needed.

On balance, I would like to invite a revision of this manuscript if you believe you can address the concerns discussed above in addition to the more specific points raised by the reviewers. I think additional data collection would be the best approach but I will not make this a requirement. It is possible that a more detailed justification for some of the interpretations of the data—potentially coupled with Bayes factors to convey the strength of evidence—may be sufficient to address the concerns.

I do not want an extended review process and will aim to make a final decision based on a revision, should you choose to submit one. As you prepare your revision, please ensure that you respond to all of the reviewer comments.

Reviewers' comments:

Reviewer's Responses to Questions

**Comments to the Author**

1. Is the manuscript technically sound, and do the data support the conclusions?

Reviewer #1: Partly

Reviewer #2: Yes

2. Has the statistical analysis been performed appropriately and rigorously?

Reviewer #1: Yes

Reviewer #2: Yes

3. Have the authors made all data underlying the findings in their manuscript fully available?

Reviewer #1: No

Reviewer #2: Yes

4. Is the manuscript presented in an intelligible fashion and written in standard English?

Reviewer #1: Yes

Reviewer #2: Yes

Reviewer #1: Review of PONE-D-255-22866 "Age differences in motivation drive performance during the sustained attention to response task"

Summary: In the current study, the authors test how age differences in motivation might explain why older adults perform better in a common sustained attention task, the SART. Using a common age-related difference experimental paradigm, younger (N = 25) and older (N = 25) older adults completed a SART that contained a speed-accuracy trade-off (SATO) titration manipulation. This manipulation was used to equate behavioral performance and put both age groups on a similar accuracy over speed bias. Following the full titration period, participants were then given an unexpected motivation manipulation for the final block of the SART. Both groups achieved similar titration levels leading to slower reaction times and increased accuracy but this change was larger for younger adults.

Evaluation: The study uses a unique approach to put older and younger adults on a level playing field by equating sustained attention performance via titration. I also appreciate the authors preregistration of the experiment and posting of the data (although it would not download for me). However, outside of that, the study only revealed nominal insights into age differences in sustained attention performance. Thus, I cannot recommend publication of the article in its current form. Below I provide some comments and concerns that I hope the authors can use to revise their manuscript for submission elsewhere.

CONCERNS:

1) In the introduction, the authors discuss longer reaction times (RTs) being a byproduct of declines in sustained attention. I do not think that is fully appropriate. This decline in RTs is more often argued as a decline in general processing speed. Surprisingly, the authors don't mention any contradictory views that what they are seeing is just a manifestation of processing speed differences. It would be good for the authors to consider alternative viewpoints to help explain their results.

2) The authors removed RTs that were < 150ms as anticipatory responses, but they did not do any sort of outlier removal for excessively long RTs. These RTs are likely impacting comparisons of RTs between the groups and should be removed or handled in some way (and noted as a deviation from the preregistration since this was not listed.

3) I am a little confused and skeptical of the analyses splitting the groups into high and low performers. Perhaps I missed it, but I did not see it in the preregistration document. If this is an exploratory analysis it should be listed as such. Additionally, was this done across the whole sample and so high performers could be younger or older adults? How does this follow from the main purpose of the paper (examining age-differences). This seems very odd to me and does not fit with the paper.

4) In general, I recommend that the authors consider adding more information around their inferential tests regarding effect sizes (e.g., Cohen's d for t-tests and confidence intervals) and Bayesian evidence (e.g., Bayes Factors). Given several null results, it it important to better understand if these effects are truly nonexistent or just weak given the sample size.

5) In several places of the paper, the authors talk about the vigilance decrement, but this study does not actually test this (at least not formally). My understanding is that there is a difference between overall vigilance (accuracy or RT in the task) and the vigilance decrement (**changes** in accuracy or RT across the task). Many of the analyses presented address age-differences in overall vigilance. The "vigilance decrement" only is addressed in the seemingly exploratory high vs. low performer analyses presented in the Titration section.

6) Given the SART often elicits false alarms (responding to no-go trials), it might be worth considered a signal detection approach to analyze the data rathe than using raw accuracy scores (which appear to be largely at ceiling). This could also more specifically address changes in different types of response biases across the age groups.

Reviewer #2: The paper focuses on an interesting question and has important results. The authors can show that in a task where older and younger adults initially perform equally well - both in terms of speed and accuracy - an additional monetary motivation affects only the younger adults, increasing their accuracy significantly, while the older adults perform in the same way as before. Thus, motivation is the main driver of age-differences here: The older adults seem to be already highly motivated without the monetary incentive, while the young participants increase their performance strongly once the monetary reward is in place.

The paper is well-written, and analyses are well-done. My only concern is about the small sample size, but the authors rely on power analyses for their decision of sample size. They comment in the limitations section on the selective sampling, which might be an issue especially with regard to the motivation - those older adults who participate in university research do have a high level of intrinsic motivation, while those students who participate in research might have a high preference for earning money. This is what shows in the experiment. But, the study does still contribute especially to methods discussions, as this difference in preferences between older and younger research participants probably holds for most studies comparing old and young samples. Thus, what the authors show here should be considered also in the interpretation of other results from other studies.

**Do you want your identity to be public for this peer review?** For information about this choice, including consent withdrawal, please see our Privacy Policy

Reviewer #1: No

Reviewer #2: No

---

## [Author Response · Author response to Decision Letter 1]

18 Sep 2025

Dear Dr Sewell,

Thank you very much for the consideration of our article for Plos1 and the reviewers for their helpful comments. Below we respond to all the comments received by yourself and both reviewers. Responses to each comment are highlighted in blue.

Journal Requirements:

Authors’ response: We have now met all the style requirements.

SH was supported by the Economic and Social Research Council Grant: ES/P000681/1.

Please state what role the funders took in the study. If the funders had no role, please state: "The funders had no role in study design, data collection and analysis, decision to publish, or preparation of the manuscript." If this statement is not correct you must amend it as needed.Please include this amended Role of Funder statement in your cover letter; we will change the online submission form on your behalf.

The funders had no role in study design, data collection and analysis, decision to publish or preparation of the manuscript.

Authors’ response: Please, see below.

Authors’ response: We have now reviewed all this text and provide here the full amended role of the funder statement:

“SH was supported by the Economic and Social Research Council Grant: ES/P000681/1. The funder had no role in the study design, data collection and analysis, decision to publish, or preparation of the manuscript. This does not alter our adherence to PLOS ONE policies on sharing data and materials.”

Authors’ response: Please, see below.

Additional requirements

3. Have the authors made all data underlying the findings in their manuscript fully available?

Reviewer #1: No

Reviewer #2: Yes

Authors’ response: The full underlying dataset has been available on osf.io since the time of submission to the journal (https://osf.io/rsyzb). In addition, the analysis scripts used to produce the findings haves now been uploaded to Github (https://github.com/SimonHanzal/Exploring_Attention/).

Academic Editor points

Thank you for submitting your manuscript to PLOS ONE. After careful consideration, we feel that it has merit but does not fully meet PLOS ONE’s publication criteria as it currently stands. Therefore, we invite you to submit a revised version of the manuscript that addresses the points raised during the review process.

Authors’ response: We thank you very much for appreciating the merits of our paper, choosing such excellent reviewers as well as reading the paper yourself making further/clarifying points, and giving us a chance to improve the paper in a resubmission.

Academic Editor points

Thank you for submitting your manuscript to PLOS ONE. I have now received reviews from two experts, who are split in their evaluations. Reviewer 1 recommends rejection on the basis of the study achieving only limited insights into age differences in sustained attention. Reviewer 2 is more positive in their evaluation and recommends accepting the manuscript. My own independent reading of the manuscript falls closer to that of Reviewer 1 and so I am rejecting the manuscript.

My overall impression is that the work makes an interesting contribution, and the use of the titration method is a very useful way of establishing functional equivalency in task performance across older and younger groups.

Authors’ response: We thank you very much for appreciating our chosen method and the interesting contribution our paper makes.

However, like Reviewer 1, I was unclear about the rationale for the median split analyses (i.e., data visualized in Figure 3).

Authors’ response: We thank you for highlighting this point. See below how we addressed it in the comment to reviewer 1 and yourself:

You are both correct that this analysis of the titration window was not pre-registered. The purpose of the analysis should have thus been more clearly outlined, and we have now done this. The only goal of this analysis was to demonstrate (as a sanity check) that the titration did indeed work in terms of performance changes in all (high and lower performing) participants, via the adaptation of the response window. We did this across the whole sample as a further age split would have underpowered this essential validation of our titration method.

Also, I think there are a few places where the relatively small sample size creates some issues in the data, power analysis notwithstanding. For example, on page 14, an interaction effect is dismissed with p = .051. I would be reluctant to completely rule out the possibility that the initially better performers sped up whereas the initially worse performers maintained a more consistent response speed over the duration of the task. In any case, there is a question of whether some of the null results are true nulls or if they reflect low power (as noted by Reviewer 1).

Authors’ response: We thank you for these recommendations. We have now added additional inferential details (Cohen’s d) and further added Bayes Factor analyses to further quantify the strength of the effects. Regarding the interaction you mention (now on page 17) in particular, after removing the excessively long RTs (see suggestion of reviewer 1) this interaction disappeared at p = .256 and a Bayesian analysis also did not support the effect, so we do not interpret it.

Perhaps calculating Bayes factors to quantify relative support for the presence/absence of an effect would be a more illuminating way to go.

Authors’ response: We fully agree and give more details (see both above and below).

Another option would be to opt for additional data collection, though there may be good reasons why this is not a viable option.

Authors’ response: We thank the editor for this suggestion but unfortunately the 1st author is now no longer in a research post and further data collection is not possible. Instead, we decided to take the other suggested approach and now provide further clarificatory Bayesian analyses for the marginal results in particular on page 14 (now page 17) and for the 2 main interactions we found for overall and no-go accuracy. For the latter we also report Cohen’s d (see below).

The biggest issue that leapt out to me, however, is the interpretation of the key data in Figure 4. In particular, the interaction showing a larger improvement in nogo accuracy in the younger group after the monetary motivation is introduced. I worry that the conclusions about much higher levels of improvement in younger participants are overstated, as (1) the younger group appears to suffer slightly poorer titrated performance and (2) it is not clear to me that incentivized performance actually differs across age groups. Admittedly, this criticism is tempered by the data showing overall accuracy, but the fact that nogo trials are much more diagnostic, I think a more detailed justification for your interpretation is needed.

Authors’ response: We very much agree that these two interactions are the crux of the paper, and we thus ran further Bayesian analyses on each of these effects. Both analyses further confirm the differential motivation effects we found (see pages 18-20).

Importantly, in addition we followed the excellent suggestion of reviewer 1 and now report d’ for accuracy in addition to our main findings regarding the motivational effects. We found that the d’ results again further underpin our main motivational finding with a significantly higher d’ in the younger (over the older) participants in the motivation condition. As these analyses deviate from our initially pre-registered approach (as outlined on page 13), we have added them (and related clarifications) at the end of the relevant result section (see page 19).

On balance, I would like to invite a revision of this manuscript if you believe you can address the concerns discussed above in addition to the more specific points raised by the reviewers. I think additional data collection would be the best approach but I will not make this a requirement. It is possible that a more detailed justification for some of the interpretations of the data—potentially coupled with Bayes factors to convey the strength of evidence—may be sufficient to address the concerns.

I do not want an extended review process and will aim to make a final decision based on a revision, should you choose to submit one. As you prepare your revision, please ensure that you respond to all of the reviewer comments.

Reviewer #1 points:

Summary: In the current study, the authors test how age differences in motivation might explain why older adults perform better in a common sustained attention task, the SART. Using a common age-related difference experimental paradigm, younger (N = 25) and older (N = 25) adults completed a SART that contained a speed-accuracy trade-off (SATO) titration manipulation. This manipulation was used to equate behavioral performance and put both age groups on a similar accuracy over speed bias. Following the full titration period, participants were then given an unexpected motivation manipulation for the final block of the SART. Both groups achieved similar titration levels leading to slower reaction times and increased accuracy but this change was larger for younger adults.

Evaluation: The study uses a unique approach to put older and younger adults on a level playing field by equating sustained attention performance via titration. I also appreciate the authors preregistration of the experiment and posting of the data (although it would not download for me). However, outside of that, the study only revealed nominal insights into age differences in sustained attention performance. Thus, I cannot recommend publication of the article in its current form. Below I provide some comments and concerns that I hope the authors can use to revise their manuscript for submission elsewhere.

CONCERNS:

1) In the introduction, the authors discuss longer reaction times (RTs) being a byproduct of declines in sustained attention. I do not think that is fully appropriate. This decline in RTs is more often argued as a decline in general processing speed. Surprisingly, the authors don't mention any contradictory views that what they are seeing is just a manifestation of processing speed differences. It would be good for the authors to consider alternative viewpoints to help explain their results.

Authors’ response: We fully agree that there are additional theoretical interpretations of longer reaction times in terms of a decline in general processing speed and its declines in older age. As per the suggestion, the introduction has been expanded to cover this explanation (page 3) but please note that the main rationale of the paper is to assess the effect of motivation on SART accuracy rather than a study of reaction time decline.

2) The authors removed RTs that were < 150ms as anticipatory responses, but they did not do any sort of outlier removal for excessively long RTs. These RTs are likely impacting comparisons of RTs between the groups and should be removed or handled in some way (and noted as a deviation from the preregistration since this was not listed.

Authors’ response: We thank the reviewer for this excellent observation. Due to the standard employed in the typical version of the SART (Robertson, 1997) and the anticipated shorter response windows (<600ms) where responses towards the very end of the trial were possible, we initially designed the task with the intent not to trim any longer RTs. On reflection this was an oversight, as these may have slightly skewed the findings pertaining to the participants with a long response window. We have now removed outliers by trimming all reaction times at participant level that surpassed 3 SDs (Vankov, 2023) from the age group mean during the experiment (as outlined on page 13) and re-run the analyses. Importantly, this adjustment had no effect on the reported effects (but see the p = 0.042 effect on page 14 which we newly addressed).

3) I am a little confused and skeptical of the analyses splitting the groups into high and low performers. Perhaps I missed it, but I did not see it in the preregistration document. If this is an exploratory analysis it should be listed as such. Additionally, was this done across the whole sample and so high performers could be younger or older adults? How does this follow from the main purpose of the paper (examining age-differences). This seems very odd to me and does not fit with the paper.

Authors’ response: The reviewer is correct that this analysis of the titration window was not pre-registered. The purpose of the analysis should have been more clearly outlined, and we have now done this (see page 15). The main goal of this analysis was to demonstrate (as a sanity check) that the titration did indeed work perfectly in terms of performance changes in all (high and lower performing) participants via the adaptation of the response window. We did indeed do this across the whole sample as a further age split would have underpowered this essential validation of our titration method.

4) In general, I recommend that the authors consider adding more information around their inferential tests regarding effect sizes (e.g., Cohen's d for t-tests and confidence intervals) and Bayesian evidence (e.g., Bayes Factors). Given several null results, it is important to better understand if these effects are truly non existent or just weak given the sample size.

Authors’ response: We thank the reviewer for these recommendations. We agree that providing additional detail to the results, especially where they are marginal, will help with the interpretation of the findings. We have thus reported additional inferential details (Cohen’s d, Cohen’s f2, generalised η², beta coefficients) and have further added Bayes Factor analyses to quantify the strength of the effects (see response to the academic editor for more detail).

5) In several places of the paper, the authors talk about the vigilance decrement, but this study does not actually test this (at least not formally). My understanding is that there is a difference between overall vigilance (accuracy or RT in the task) and the vigilance decrement (**changes** in accuracy or RT across the task). Many of the analyses presented address age-differences in overall vigilance. The "vigilance decrement" only is addressed in the seemingly exploratory high vs. low performer analyses presented in the Titration section.

Authors’ response: The reviewer is correct, and we apologise for the poor clarity: although our results have implications for the vigilance decrement, we test age differences in overall vigilance. We have now changed the terminology as per the reviewer proposal when referring to our findings, while retaining reference to the vigilance decrement where relevant (page 3 and pages 26-27 ).

6) G

---

## [Decision Letter · Decision Letter 1]

30 Oct 2025

Age differences in motivation drive performance during the sustained attention to response task

PONE-D-25-22866R1

Dear Dr. Hanzal,

We’re pleased to inform you that your manuscript has been judged scientifically suitable for publication and will be formally accepted for publication once it meets all outstanding technical requirements.

Kind regards,

David Keisuke Sewell, Ph.D.

Academic Editor

PLOS ONE

Additional Editor Comments (optional):

Dear Dr Hanzal,

Thank you for submitting your revised manuscript to PLOS ONE. Since the original Reviewer 2 recommended acceptance of your original submission, I felt comfortable basing a decision on your revised manuscript on a review from the original Reviewer 1, complemented by my own reading of the manuscript. I appreciate the attention in addressing the reviewers' comments and particularly for the enhanced reporting of statistical results that have provided further clarity on the findings. I agree with the reviewer that all of the concerns have been adequately addressed and am very pleased to accept your manuscript for publication.

Best regards,

Dr David Sewell

Reviewers' comments:

Reviewer's Responses to Questions

**Comments to the Author**

Reviewer #1: All comments have been addressed

2. Is the manuscript technically sound, and do the data support the conclusions?

Reviewer #1: (No Response)

3. Has the statistical analysis been performed appropriately and rigorously?

Reviewer #1: (No Response)

4. Have the authors made all data underlying the findings in their manuscript fully available?

Reviewer #1: (No Response)

5. Is the manuscript presented in an intelligible fashion and written in standard English?

Reviewer #1: (No Response)

Reviewer #1: The authors have address my major concerns. I appreciate their work on the revision and improved clarification.

**Do you want your identity to be public for this peer review?** For information about this choice, including consent withdrawal, please see our Privacy Policy

Reviewer #1: No

---

## [Editor Report · Acceptance letter]

PONE-D-25-22866R1

PLOS ONE

Dear Dr. Hanzal,

I'm pleased to inform you that your manuscript has been deemed suitable for publication in PLOS ONE. Congratulations! Your manuscript is now being handed over to our production team.

Kind regards,

on behalf of

Dr. David Keisuke Sewell

Academic Editor

PLOS ONE